# Identification and Characterization of the Gene Responsible for the O^3^ Mating Type Substance in *Paramecium caudatum*

**DOI:** 10.3390/microorganisms12030588

**Published:** 2024-03-15

**Authors:** Yuta Chiba, Yasuhiro Takenaka, Nobuyuki Haga

**Affiliations:** 1Department of Biological Sciences, Faculty of Science and Technology, Senshu University of Ishinomaki, Miyagi 986-8580, Japan; camellia-drops.1833@softbank.ne.jp; 2Department of Bioregulatory Science, Graduate School of Medicine, Nippon Medical School, Tokyo 113-8602, Japan; yasuhiro-takenaka@nms.ac.jp

**Keywords:** mating-type substance, protein kinase, EF hand motif, phylogenic tree, RNA interference, silent mutations, microinjection, *Paramecium caudatum*

## Abstract

The process of sexual reproduction in eukaryotes starts when gametes from two different sexes encounter each other. *Paramecium*, a unicellular eukaryote, undergoes conjugation and uses a gametic nucleus to enter the sexual reproductive process. The molecules responsible for recognizing mating partners, hypothetically called mating-type substances, are still unclear. We have identified an O^3^-type mating substance polypeptide and its gene sequence using protein chemistry, molecular genetics, immunofluorescence, RNA interference, and microinjection. The O^3^-type substance is a polypeptide found in the ciliary membranes, located from the head to the ventral side of cells. The O^3^-type substance has a kinase-like domain in its N-terminal part located outside the cell and four EF-hand motifs that bind calcium ions in its C-terminal part located inside the cell. RNA interference and immunofluorescence revealed that this polypeptide positively correlated with the expression of mating reactivity. Microinjection of an expression vector incorporating the O^3^Pc-MSP gene (*Oms3*) induced additional O^3^ mating type in the recipient clones of different mating types or syngen. Phylogenetic analysis indicates that this gene is widely present in eukaryotes and exhibits high homology among closely related species. The O^3^Pc-MSP (*Oms3*) gene had nine silent mutations compared to the complementary mating type of the E^3^ homologue gene.

## 1. Introduction

Sexual reproduction is a process of creating new offspring with a unique genome makeup that differs from their parents. This involves two parents of opposite sexes producing haploid nuclei through meiosis. The genomes of these nuclei then combine to form a diploid nucleus within a single cell through fertilization. In 1946, C.B. Metz proposed a definition for mating-type substances [1]. He defined them as “substances or molecular structures that facilitate binding reactions at the cell surface”. The conjugation process in ciliates can be divided into various phases, including the early stages of sexual cell recognition, cellular contact, and membrane fusion. Each of these phases has distinct characteristics in different species. The determination of mating type and inheritance pattern varies from species to species and is governed by species-specific molecules or rules. References for each species include *Paramecium tetraurelia* [2], *Tetrahymena thermophila* [3], *Blepharisma japonicum* [4], and *Euplotes raikovi* [5].

Research on sexual reproduction in protists started in 1937 when T.M. Sonneborn discovered a mating reaction between cells with complementary mating types in the ciliate species *Paramecium aurelia* complex [6]. In the *Paramecium aurelia* species complex, there are three mechanisms of mating-type determination: maternal inheritance, stochastic determination, and Mendelian inheritance [7]. In 1968, Hiwatashi discovered a similar phenomenon in *Paramecium caudatum* and analyzed the inheritance of mating types [8]. The mating type of *P. caudatum* is determined by a pair of alleles, which follow Mendel’s law and are passed down to offspring. Individuals with at least one dominant gene (*Mt*) express even-type (E type), and homozygotes for a recessive gene (*mt*) express odd-type (O type) [8,9]. Mating occurs between the complementary mating types of E and O within the same syngen. Sixteen syngen have been reported in *P. caudatum* [10,11]. Other members of the genus *Paramecium* have been reported as cryptic species (in some cases called syngen) [12,13,14].

When *Paramecium* finds bacteria in its environment, it consumes them and undergoes asexual reproduction, leading to an exponential population increase. During this stage, *Paramecium* does not show any mating reactivity. However, when the bacteria in the environment are exhausted, feeding stops, and adult *Paramecium* expresses mating reactivity. When E-type and O-type paramecia expressing mating reactivity come into contact, they adhere using cilia on their ventral sides and trigger a mating reaction [15]. After the mating reaction, the first notable change is that the micronucleus, known as the germ nucleus, pops out of the pocket-like depression of the macronucleus within 10 min. This process is called early micronuclei migration (EMM) [16]. EMM is associated with increased intracellular calcium ion concentration [17]. 

Approximately 45 min after the start of the mating reaction, the cilia located at the front end of the cell begin to degenerate [18]. This leads to the adhesion between the cells occurring at the cell membrane, called the holdfast union. Approximately 90 min after the start of the mating reaction, the cilia near the oral apparatus degenerate, and the region where the two cells contact expands to form a paroral union. During the paroral-union stage, the cell membrane fuses in a specific part of the oral region, exchanging the germ nuclei. Following this, both cells of a mating pair independently combine their germ nucleus with that of their partner cell, forming a new generation of genomes within the synkaryon [19]. 

Mating type specificity is only necessary when mating begins. Two individuals of the same mating type can mate with each other. This can be achieved through chemical induction of conjugation methods, which allow the mating process to proceed normally, even between individuals of the same mating type [20]. Mating pairs can also be formed between different species (interspecific mating pairs) of the *Paramecium* genus [21,22]. 

Since the mid-1950s, various studies have been conducted on mating reactions, and their key findings are summarized as follows: (1) The mating reaction occurs through cilia that grow from the front end of the cell towards the ventral region where the oral apparatus is located [18]. (2) Ciliary membrane-intrinsic proteins are crucial in the mating reaction [23,24]. (3) Potassium ions sustain the expression of mating reactivity [25]. However, the molecules corresponding to mating-type substances and integral ciliary membrane proteins involved in mating reactions have yet to be identified.

In this study, as an experimental working hypothesis, we defined O^3^ mating-type substances in *Paramecium* as substances that combine with E^3^-type cells to form cell aggregates and cause mating reactions. We demonstrate the molecule and gene that distinguish individuals of the same species with complementary mating types. Our findings support the existence of molecules postulated by C.B. Metz as mating-type substances. We discuss findings related to the gene that defines the odd-mating type of *Paramecium* belonging to Syngen 3 (O^3^Pc-MSP (*Oms3*) gene described in the study). We also discuss the feature of the catalytic domain deduced from the amino acid sequence of the mating type substance and the characteristics of the phylogenetic tree. We explore the characteristics and potential evolutionary roles of nine silent mutations that were revealed through comparison with homologous sequences of the complementary mating type E^3^. A key finding from the phylogenetic tree is that the O^3^Pc-MSP (*Oms3*) gene and its homologs were widely distributed throughout eukaryotic evolution and have been conserved in multicellular organisms.

## 2. Materials and Methods

### 2.1. Strains and Culture Methods

The strains of *Paramecium caudatum* used in the study were TAZ0460 (O^3^ type, Syngen 3) and TAZ0462 (E^3^ type, Syngen 3). These are the progeny of the KNZ series collected in Kanazawa, Japan (kindly supplied by Dr. H. Endoh, Kanazawa University). The culture medium was prepared according to Hiwatashi’s method [8]. Cells were cultured in 1.25% fresh lettuce juice diluted with Dryl’s solution [26], modified by substituting KH_2_PO_4_ for NaH_2_PO_4_ (named K-DS) at 25 °C. The culture medium was inoculated with *Klebsiella pneumonia* and cultured at 25 °C the day before use.

*Paramecium* expressing mating reactivity was obtained using the following culture method. On the first day of culture, 1 mL of the cell suspension containing several hundred paramecia was transferred to a sterile test tube, and 2 mL of fresh culture medium was added. After that, 4 mL, 8 mL, and 8 mL of new culture medium were added sequentially for three consecutive days. One day after the last feeding, cells with high mating reactivity were obtained. *Paramecium*, in which mating reactivity was not expressed, was prepared using the following method. *Paramecium* suspension that entered the stationary phase on day one from the growth phase (approximately 2.0 × 10^3^ cells) was added with 20 mL of culture medium and incubated at 25 °C for 5 h. The cells of paramecia at this stage were in the exponential growth phase and did not show mating reactivity. 

### 2.2. Preparation of Ciliary Membrane Fraction

Mass culture of paramecia was carried out by placing 1.5 L of culture solution in a 2 L Erlenmeyer flask. A suspension of paramecia (approximately 1.0 × 10^5^ cells) with a high mating reactivity was added and cultured at 25 °C for six days. It was confirmed that mating reactivity was expressed on the seventh day. Mass culture of paramecia that does not express mating reactivity was carried out using a suspension of paramecia (approximately 1.0 × 10^6^ cells) added to 2.0 L of culture medium and cultured at 25 °C for 12 h. It was confirmed that no mating reactivity was expressed.

### 2.3. Amplification of the O^3^Pc-MSP cDNA Fragment by RACE and RT-PCR 

To determine the base sequence of the O^3^Pc-MSP mRNA, cDNA fragments derived from the O^3^Pc-MSP mRNA were amplified using the 3′-RACE (rapid amplification of cDNA ends) method, 5′-RACE method, and RT-PCR (Reverse transcription-polymerase chain reaction) method. *Paramecium* cells expressing mating reactivity were collected using a hand centrifuge, and the cell density was adjusted to 1.0 × 10^3^ cells/mL. Total RNA was extracted from this cell suspension using NucleoSpin RNA XS (Takara Bio, Kusatsu, Siga, Japan). The nucleotide sequence at the 3′ end of the O^3^Pc-MSP mRNA was amplified using the 3′-RACE method (The 3′-RACE System for Rapid Amplification of cDNA Ends (Life Technologies, Carlsbad, CA, USA). The nucleotide sequence of the 3′-RACE product of the mRNA was determined using primer MSP-868L(O^3^Pc-MSP) (CCCAAGAAGAGACCATCAGC), which was synthesized from the partial amino acid sequence EFTSLSFEAQNLIK determined via mass spectrometry. The PCR reaction was performed with the following program: (94 °C–3 min, (94 °C–45 s, gradient (52–63 °C)–25 s, 72 °C–3 min) (30 cycles), 72 °C–10 min) using MSP-868L and AUAP (Life Technologies, USA). A gradient PCR thermal cycler, Dice TP600 (Takara Bio, Japan), was used for the PCR reaction. 5′-RACE was performed to amplify the base sequence of the 5′ end of the mating substance mRNA (5′-RACE System for Rapid Amplification of cDNA Ends, version 2.0 (Life Technologies, USA)). The primers used for 5′-RACE were MSP-988R (O^3^Pc-MSP) (CTGCATAATGGCTGCTTCA) and AUAP (PCR with the following protocol program: (94 °C–3 min, (94 °C–1 min, gradient (52–63 °C)–30 s, 72 °C–90 s) (30 cycles), 72 °C–10 min)).

### 2.4. Amplification of Full-Length O^3^Pc-MSP DNA by Genomic PCR Method

To amplify the entire length of the O^3^Pc-MSP gene, we extracted genomic DNA from the O^3^ *Paramecium* suspension, which was adjusted to 1.0 × 10^3^ cells/mL. We used two types of primers for PCR. MSP-1470R primer (TCATTTTTTTGCAGTTTATAAGAGC) was created from a 25-base sequence that contains the stop codon (TGA) at the 3′ end of the 3′-RACE products. MSP-1L primer (ATGGGTGCTTGTGGTGGTAAATCGG) was created from a 25-base sequence that contains the start codon (ATG) at the 5′ end of the 5′-RACE products. The PCR reaction enzyme was Titanium Taq DNA polymerase (Clontech, TaKaRa Bio, Kusatsu, Siga, Japan), and the following program was used. This step takes 2 min and involves 30 cycles of heating and cooling. The heating step was performed at 95 °C for 30 s, while the cooling step was performed at 72 °C for 10 min (95 °C–10 min, 95 °C–15 s, gradient (52~63 °C)–30 s, 72 °C–2 min) (30 cycles), 72 °C–10 min).

### 2.5. Amplification of Entire Length Mating Substance mRNA

To amplify the entire length of O^3^Pc-MSP mRNA, we used two primers: MSP-1L and MSP-1470R. We used SuperScript to synthesize and amplify cDNA derived from full-length O^3^Pc-MSP mRNA. To perform the OneStep RT-PCR System (Life Technologies, USA), we followed this program: 55 °C for 30 min, 94 °C for 2 min, 94 °C for 15 s, gradient (52–63 °C) for 30 s, 68 °C for 90 s (40 cycles), and 68 °C for 5 min.

### 2.6. Base Sequence Analysis

#### 2.6.1. Purification of 3′-RACE, 5′-RACE, RT-PCR, and PCR Products

Electrophoresis of 3′-RACE, 5′-RACE, RT-PCR, and PCR products was performed using 1% agarose (Promega, Madison, WI, USA). Submarine electrophoresis was performed at 100 V for 30 minutes using i-Mupid (Advance, Tokyo, Japan). A 0.5% TAE buffer was used to prepare the agarose plate and electrophoresis. After electrophoresis, the agarose plates were stained with ethidium bromide (Nippon Gene, Tokyo, Japan). Target DNA bands were visually searched on a transilluminator and excised from the agarose plate. Next, DNA was purified from the excised DNA band using Mini Elute gel Extraction (QIAGEN, Redwood City, CA, USA).

#### 2.6.2. Integration of Purified Product into Sequencing Vector and Amplification Using *Escherichia coli*

The purified DNA fragment was inserted into circular DNA (pCR4 vector) for sequencing using a cloning kit (TOPO TA Cloning Kit for Sequencing, Life Technologies, USA) and introduced into *E. coli* (TOP10 competent cells, Life Technologies, USA). For the selection of transformed TOP10 cells, LAMP agar medium (LB BROTH BASE) (Life Technologies, USA), 2% agar powder (FUJIFILM Wako Pure Chemical Industries, Osaka City, Osaka, Japan), and 50 μg/mL carbenicillin sodium (FUJIFILM Wako Pure Chemical Industries, Ltd.)) were used. After culturing the gene-transfected TOP10 on a LAMP agar medium at 37 °C for 18 h, 32 colonies were randomly selected from the former colonies. Colony PCR was performed to select clones transformed with the pCR4 vector. Colony PCR used a portion of *E. coli* from each of the 32 selected colonies as a direct template, and PCR was performed using the following program using the same primers as those used for amplification of the product ((95 °C–10 min, (95 °C–15 s, 57 °C–30 s, 72 °C–2 min) (30 cycles), 72 °C–10 min)). The remaining *E. coli* was inoculated onto a new LAMP agar medium and cultured at 37 °C for 18 h.

#### 2.6.3. Amplification by RCA Method and Fluorescent Labeling by Cycle Sequence Method

After selecting *E. coli* TOP10 colonies transformed with the pCR4 vector by PCR, the pCR4 vector was amplified using the Illustra TempliPhi DNA Amplification Kit (GE Healthcare, Chicago, IL, USA) through the rolling circle amplification (RCA) method. The RCA reaction was carried out using a program of 30 °C for 18 h and 65 °C for 10 min. Following the RCA reaction, the amplified RCA product was used as a template for cycle sequencing, where fluorescently labeled dNTP was bound to the desired PCR product. BigDye Terminator v3.1 Cycle Sequencing Kit (Applied Biosystems, Waltham, MA, USA) was used for the cycle sequencing method. Primer was M13 Forward (−20) Primer (Life Technologies, USA) and M13 Reverse Primer (Life Technologies, USA) were used under the following conditions (96 °C–1min, (96 °C–10 s, 55 °C–5 s, 60 °C–4 min) (30 cycles)). After the cycle sequencing reaction, Illustra AutoSeq G-50 Dye Terminator Free dNTPs were removed using a removal kit (GE Healthcare, USA), and the samples were used for nucleotide sequence analysis.

#### 2.6.4. Base Sequence Analysis

The base sequence of the sample was analyzed using ABI PRISM 3100 Genetic Analyzer (Applied Biosystems, USA). To translate a base sequence into an amino acid sequence, the website ORF finder (Genetic codes 6 Ciliate, Dasycladacean, and Hexamita Nuclear (http://www.ncbi.nlm.nih.gov/projects/gorf/, accessed on 5 January 2024)) were used. Estimating molecular weight from the complete amino acid sequence is available on the ExPASy Bioinformatics Resource Portal (http://web.expasy.org/compute_pi/, accessed on 5 January 2024), and homology searches were performed using BLAST searches.

### 2.7. Molecular Phylogeny of O^3^Pc-MSP

A multiple sequence alignment of the full-length amino acid sequence of O^3^Pc-MSP and 14 homologues was conducted using the ClustalW algorithm with default parameters by MacVector software (ver13.5.2). The maximum likelihood (ML) consensus tree for O^3^Pc-MSP was generated from bootstrap analysis with 1000 replicates of O^3^Pc-MSP and 14 homologues based on the ClustalW amino acid sequence alignment with Le_Gascuel_2008 model [27] with a discrete Gamma distribution (+G) model by MEGA6.06 [28].

### 2.8. Preparation of Anti O^3^PC-MSP Polyclonal Antibody

Sequences suitable for antigenic determinants were selected from the mating substance’s estimated amino acid sequence. The following two points were established as conditions for selecting antigenic determinants: (1) A highly unique partial amino acid sequence; (2) A sequence with a C-terminal cysteine residue that provides a binding site for antigenic stimulation carrier protein KLH (Keyhole limpet hemocyanin). A synthetic peptide was produced as an antigenic determinant based on the partial amino acid sequence KSDSKEQTKTKEQKQC from the 7th to the 21st. A synthetic peptide with this amino acid sequence was used as an antigenic determinant for a polyclonal antibody. The production of synthetic peptides, rabbit immunization, whole blood collection, and antiserum production was outsourced (Scrum Co., Ltd. Tokyo, Japan). Synthetic peptides serving as antigenic determinants were injected into two rabbits, whole blood was collected 63 days later, and antiserum was obtained using an affinity column purification method.

### 2.9. Preparation of the Ciliary Membrane Intrinsic Protein Fraction 

A protein fraction specific to the ciliary membrane was prepared using a modified Yano method [29]. After adjusting the cell density of *Paramecium* to 4.0 × 10^4^ cells/mL, Dryl’s solution and STEN solution (20 mM Tris-HCl, 6 mM NaCl, 2 mM 2Na_-_EDTA, 500 mM sucrose, pH 7.5) were mixed in a 1:1 ratio. Then, four volumes of the mixture were added to the cell suspension and incubated on ice for 30 min. After that, 180 mM KCl and 60 mM CaCl_2_ were added to 30 mM and 10 mM final concentrations, respectively. Then, the mixture was incubated on ice for 10 min. Next, the sample was centrifuged at 1000× *g* for 4 min at 4 °C using a low-temperature centrifuge (MX-160 (TOMY, Tokyo, Japan)), and the supernatant was collected. The supernatant was centrifuged at 15,000× *g* for 40 min at 4 °C, and the precipitated cilia were suspended in TE buffer (1 mM Tris-HCL, 0.1 mM EDTA, pH 8.3) and vortexed at 4 °C for 5 min. This operation separated the ciliary membrane and axoneme. Next, the ciliary membrane and axoneme suspension were separated using an ultracentrifuge (CP70MX, Hitachi, Tokyo, Japan). Ultracentrifugation was performed at 48,000× *g* for 30 min at 4 °C. The precipitate was resuspended in 300 μL of Tris buffer (10 mM Tris-HCl, pH 8.0). Sucrose density gradient centrifugation was used to obtain only the ciliary membrane components from the resuspension. The resuspension was placed on top of a density gradient solution made with the Tris-HCL buffer containing 20%, 45%, and 66% (*w*/*v*) sucrose and ultracentrifuged at 208,000× *g* for 90 min at 4 °C. After ultracentrifugation, ciliary membrane components were collected from a border of the Tris buffer containing 20% and 45% sucrose. After collection, the ciliary membrane components were mixed with a 10-fold volume of the Tris buffer and then ultracentrifuged at 48,000× *g* for 30 min at 4 °C. The precipitate derived from ciliary membrane components was suspended in 100 μL of TBS buffer (20 mM Tris-HCl, 150 mM NaCl, pH 7.5). This was used as the ciliary membrane fraction. 

The acetone precipitation method concentrated the protein components in the ciliary membrane fraction. Ice-cold acetone (Wako Pure Chemical, Japan), was added to the ciliary membrane fraction in a 1:1 (*v*/*v*) ratio and incubated at 4 °C for 10 min. The mixture was then centrifuged at 4 °C and 14,000× *g* for 10 min to precipitate the proteins in the fraction. This process was repeated three times after removing the acetone. Finally, the precipitated protein was dried and dissolved in sterile ultrapure water.

A portion of the ciliary membrane fraction obtained via the above method was taken out, diluted 10 times with sterile ultrapure water, and used for protein quantification. A DC Protein Assay kit (BIO-RAD Laboratories, Tokyo, Japan) was used for quantification. Absorbance was measured using UVmini-1240 (SHIMADZU, Kyoto, Japan).

### 2.10. Detection of O^3^Pc-MSP Polypeptides by Western Blotting

Samples for polyacrylamide gel electrophoresis were prepared as follows. LDS sample buffer (424 mM Tris-HCl, 4.36 M glycerol, 564 mM Tris base, 292 mM LDS, 2.04 mM EDTA, 0.88 mM Serva Blue G250, 0.7 mM phenol red, pH 8.5 (Life Technologies, USA) and NuPAGE reducing agent (0.5 M DTT) (Life Technology, USA) were added to ciliary membrane fraction (2.5 mg/mL protein), mixed, and heated at about 90 °C for 10 min. The samples were then left on ice for 10 min. Electrophoresis was performed using a NuPAGE 10% Bis-Tris gel (Life Technologies, USA) in an electrophoresis buffer containing MOPS SDS Running buffer and NuPAGE antioxidant at 200 volts for 45 min. The proteins separated through electrophoresis were transferred to a PVDF membrane (iBlot Gel Transfer Stacks PVDF, Mini) (Life Technologies, USA)) using an iBlotTM Gel Transfer Device (Life Technologies, USA). To detect the O^3^Pc-MSP, we used an anti-MSP antibody and the Western Breeze Chromogenic Western Blot Immunodetection kit (Life Technologies, USA).

### 2.11. Indirect Immunofluorescence

For indirect immunofluorescence, an equal volume of K-PB (25 mM KCl, 2 mM phosphate buffer, pH 7.2 containing 0.01% Triton X-100 and 8% formaldehyde) was added to the *Paramecium* suspension (4.0 × 10^3^ cells/mL) and fixed on ice for 20 min. After fixation, the cells were washed thrice with K-PB. Then, they were reacted with the anti-MSP antibody diluted 250 times with 2 mM phosphate buffer (pH 7.2) for 2 h on ice. After the reaction, the sample was washed thrice with K-PB. Then, a secondary antibody, Alexa Fluor-488 conjugated goat anti-rabbit IgG (F (ab) fragment (Life Technologies, USA), was diluted 400 times with 2 mM phosphate buffer (pH 7.2) and added to the sample. The reaction took place on ice for 2 h. After washing the sample three times with K-PB, 10 μL was transferred onto a glass slide. The sample was then mounted with VECTASHIELD (Vector Laboratories, Newark, CA, USA) and observed under fluorescence microscopy ECLIPSE Ni (Nikon, Tokyo, Japan).

### 2.12. Measuring the Intensity of Fluorescence around the Cell Membrane

The fluorescence intensity of indirect fluorescence images was measured using the NIS element (Nikon, Japan). Ten cells were randomly selected and imaged under a fluorescence microscope with a shutter speed of 100 ms and an analog gain of 1.2 db. The fluorescence intensity of the image was measured quantitatively. To do this, 60 spots (each measuring 200 μm^2^) were placed around the cell membrane, following the outer shape of the cell, using the bright field image as a reference. These spots were then copied and applied to the fluorescence image, and the fluorescence intensity within each spot was measured. This allowed us to calculate the fluorescence intensity per unit area. Fluorescence intensity was measured in 10 cells and subjected to Tukey’s multiple comparison test to determine significant differences.

### 2.13. Detection of O^3^Pc-MSP mRNA by Semi-Quantitative RT-PCR Method

After washing O^3^
*Paramecium* cells three times with K-DS, 50 cells were randomly selected using a binocular stereomicroscope. These 50 cells were then transferred to 100 μL of K-DS. Total RNA was extracted from the selected cells using NucleoSpin RNA XS (TaKaRa Bio, Japan), and this RNA was used as a template. Using the SuperScript III OneStep RT-PCR System (Life Technologies, USA), we determined the expression level of the O^3^Pc-MSP mRNA from extracted total RNA using two types of PCR primers: MSP-431L (CAGCTGCCGACTACATGAAA) and MSP-1007R (CTGCATAATGGCTGCTTCCA) through semi-quantitative RT-PCR. The RT-PCR reaction was carried out using the following program: 55 °C for 30 min, 94 °C for 2 min, followed by 40 cycles of 94 °C for 15 s, 57 °C for 30 s, and 68 °C for 60 s, and a final extension step of 68 °C for 5 min. As a control, the expression level of α-tubulin mRNA was examined using α-tube-rt-lp1 (ACAAAGGCTCTCTTGGCATACATA) and Pc-tube-up1 (GCAACAATCAAGACAAAGAGAACC) primers.

### 2.14. Preparation of Paramecium at the Beginning of the Mating Process

O^3^- and E^3^-type cells expressing mating reactivity were mixed in a 1:1 ratio to initiate the mating reaction. After 20 min, the cell aggregates were treated with a K-PB solution containing 0.005% Triton X-100 and 4% formaldehyde to fix the specimens. 

### 2.15. Preparation of Double-Stranded RNA Synthetic Plasmid Vector

We created a Pc-MSP knockdown strain using feeding RNA interference by designing a plasmid vector that can synthesize double-stranded RNA in *E. coli*. The target sequence was the entire length of O^3^Pc-MSP mRNA (1470 bp) and two regions: 154-93 and 1003-1254. A DNA fragment with recognition sequences for restriction enzymes *Bgl* II and *Kpn* I was added to the target base sequence to insert the sequence into a double-stranded RNA expression vector. The regions of the base sequence 154-393 and 1003-1254 were amplified via genomic PCR using MSP-154L-bgl2 (AGATCTCTTGGAGAGGGTTCCTATGGTT) and MSP-393R-kpn1 (GGTACCGAGCAATTCTCCGCCATTTA) as primers, and MSP-1003L-bgl2 (AGATCTTGCAGTTGATTGCAGGGTAA) and MSP-1354R-kpn1 (GGTACCCTTTGCTGAAGCCACAAGAA) as primers, respectively. The full-length O^3^Pc-MSP cDNA was introduced into a pCR4 vector, which was used as a template. The PCR reaction was performed using Titanium Taq DNA polymerase (Clontech/TaKaRa Bio, Japan) and the following program: 95 °C–10 min, 95 °C–15 s, gradient (52~63 °C)–30 s, 72 °C–2 min (30 cycles), and 72 °C–10 min.

After the completion of PCR, TOP10 strains were selected based on the nucleotide sequence of the PCR product. The selected strains showed a PCR product that completely matched the target nucleotide sequence of O^3^Pc-MSP mRNA. Colonies of the selected TOP10 strains were then cultured in a LAMP liquid medium containing LB BROTH BASE (Life Technologies, USA) and 50 μg/mL carbenicillin sodium (Wako Pure Chemical, Japan). The shaking culture was performed at 37 °C for about 18 h using the PERSONAL-11 reciprocating shaker (TAITEC, Koshigaya, Saitama, Japan). To prepare the TOP10 strains, they were cultured by shaking and then subjected to miniprep using the QIAprep spin miniprep kit (QIAGEN, USA). Next, a plasmid was created by cloning a PCR product, to which recognition sequences for restriction enzymes *Bgl*Ⅱ and *Kpn*Ⅰ were added to the target base sequence. Finally, the extracted plasmid was obtained.

### 2.16. Insertion of Target Base Sequence into L4440 Vector

The PCR product underwent treatment with restriction enzymes *Bgl*II and *Kpn*I to extract only the desired base sequence. The product was digested at 37 °C for 2 h. Next, 2% agarose submarine electrophoresis was performed using an agarose plate. The agarose plate was stained with ethidium bromide to visualize the DNA fragments. After staining, a band of the target size (approximately 250 bp) was cut out on a transilluminator. The target band was purified via Mini Elute Gel Extraction (QIAGEN, USA). The region from base pair 154-393 was defined as the 5′ fragment and the region from base pair 1003-1254 was defined as the 3′ fragment. To express double-stranded RNA, we used the L4440 vector (kindly supplied by Dr. Manabu Hori (Yamaguchi University, Yamaguchi, Japan), which has a T7 promoter upstream and downstream of the multiple cloning site, along with restriction enzymes. After linearizing the vector through digestion with *Bgl*II and *Kpn*I, we conducted submarine electrophoresis using a 1% agarose plate. The agarose plate was stained with ethidium bromide, and we cut out a DNA fragment of the desired size (approximately 2700 bp) before using Mini Elute Gel Extraction (QIAGEN, USA) to purify the target band (L4440 fragment). The ends of each fragment were ligated with two combinations of purified L4440 fragment and 5′ fragment, or L4440 fragment and 3′ fragment. To create two combinations of purified L4440 fragments, the 5′ and 3′ fragments were ligated at the ends of each fragment. The ligation process involved the addition of 2× Rapid Ligation Buffer (Promega, Tokyo, Japan) and T4 DNA Ligase to the mixture of the L4440 fragment and either the 5′ or 3′ fragment (Promega, Japan). The mixture was left at room temperature for 30 min and incubated at 15 °C for approximately 24 h. The ligated plasmid was purified using a Mini Elute Cleanup kit (QIAGEN, USA). The resulting ligation of the L4440 fragment and the 5′ fragment was named the 5′-L4440 vector, and the ligation of the L4440 fragment and the 3′ fragment was named the 3′-L4440 vector. The 5′- or 3′-L4440 vector was transformed into *E. coli* HT115 strain (kindly supplied by Dr. Manabu Hori). As a control, the L4440 vector in which the cDNA fragment derived from Pc-MSP mRNA was not inserted was transformed into *E. coli* HT115 strain. To select the transformed strain HT115, LB/AT agar medium containing 100 μg/mL carbenicillin sodium and 12.5 μg/mL tetracycline hydrochloride (Wako Pure Chemical, Japan) was used. After culturing on LB/AT agar medium at 37 °C for approximately 18 h, 16 colonies were randomly selected from each plate, and the transformed clones of each vector were selected through colony PCR. The primer sequence is TAATACGACTCACTATAGGG (T7 promoter). PCR program: 95 °C–10 min, 30 cycles of 95 °C–15 s, 61 °C–30 s, 72 °C–2 min, followed by 72 °C–10 min. Afterward, the base sequence of the DNA band in the transformed colony was determined. The primers used for cycle sequencing were M13 Forward (−20) Primer (Life Technologies, USA) and L4440 seqF (AGCGAGTCAGTGAGCGAG).

### 2.17. RNA Interference Method through Feeding

*Paramecium* was fed the *E. coli* HT115 strain carrying 5′-L4440 and 3′-L4440 vectors and the *E. coli* HT115 strain carrying the control L4440 vector [30]. Knockdown of *Paramecium* O^3^Pc-MSP mRNA was induced using the following steps. 

#### 2.17.1. The Procedure of *E. coli* Suspension for Feeding

We selected individual colonies from *E. coli* strains that were transformed with either 5′- or 3′-L4440 vectors and strains that were transformed with control L4440 vectors. We then incubated them overnight at 37 °C in 2 mL of LB/AT medium with shaking. Next, we added 100 μL of each strain to 5 ml of LB/AT medium and shook them at 37 °C for 2 h. After that, we induced the expression of double-stranded RNA by adding IPTG (Isopropyl β-D-1-thiogalactopyranoside) to each culture solution to a final concentration of 0.4 mM and shook them at 37 °C for 3 h. We centrifuged the *E. coli* cells at 4000 rpm for 10 min and suspended them in 10 mL of K-DS containing 1.25% (*w*/*v*) lettuce juice. We measured the turbidity of the *E. coli* suspensions by checking the optical density (OD) at 600 nm using a spectrophotometer (UVmini-1240 (SHIMADZU, Japan)). After measuring the OD, we mixed the *E. coli* suspension with K-DS containing 1.25% (*w*/*v*) lettuce juice until the OD_600_ was 0.125. We mixed well carbenicillin sodium (final concentration, 100 μg/mL) and IPTG (final concentration, 0.4 mM). The resulting mixtures were used as the *E. coli* suspensions for feeding. 

#### 2.17.2. Induction of O^3^Pc-MSP Knockdown by Feeding RNA Interference

The O^3^-type *Paramecium*, which causes knockdown, was allowed to swim in the K-DS solution for approximately three hours to release the food vacuole contents outside the cells. After that, it was washed three times with K-DS solution. The *E. coli* suspensions were added to the *Paramecium* suspensions for RNA interference. The cell density was maintained at 400 cells/mL. The knockdown-treated strains were cultured at 25 °C for approximately 45 h. The strains fed with the HT115 carrying the 5′-L4440 vector were designated as 5-KD (knockdown on the 5′ ends). Those fed the HT115 strain carrying the 3′-L4440 vector were designated as 3-KD (knockdown on the 3′ ends). The strains fed with the HT115 strain carrying the L4440 vector were designated as controls. To confirm the knockdown effect of O^3^Pc-MSP mRNA, we compared the expression level of the O^3^Pc-MSP mRNA among three types of cell samples: 5-KD, 3-KD, and control. In addition, the intracellular localization of O^3^Pc-MSP was compared using indirect immunofluorescence.

#### 2.17.3. Calculation of the Percentage of Cells Expressing Mating Reactivity

The experiment involved calculating the percentage of cells displaying mating reaction activity in three O^3^Pc-MSP knockdown-treated lines. For this, thirty cells were randomly selected from each cell sample and placed in individual wells of a 96-well plate. Next, E^3^-type *Paramecium*, consisting of approximately 100 cells expressing mating reaction activity, was added to each well. After standing at room temperature for 10 min, the formation of cell aggregates due to the mating reaction was examined. This process was repeated three times, and the average value and standard deviation of the percentage of cells displaying mating reactivity were calculated.

#### 2.17.4. Generation of O^3^Pc-MSP Knockdown Recovery Strain

The following procedure was used to prepare the *E. coli* suspension for knockdown recovery experiments. Firstly, a single colony was taken from the transformed *E. coli* HT115 strain and cultured at 37 °C overnight with shaking in 2 mL of LB/AT medium. Then, 100 μL of the *E. coli* suspension was added to 5 mL of LB/AT medium and cultured with shaking at 37 °C for 5 h. It is worth noting that IPTG was not added, and double-stranded RNA was not expressed in this experiment. After the shaking culture, the *E. coli* bacteria were collected by centrifugation at 4000 rpm for 10 min and suspended in 10 mL of 1.25% (*w*/*v*) lettuce juice K-DS. Next, after measuring the turbidity of the HT115 suspension, the suspension was diluted with K-DS containing 1.25% (*w*/*v*) lettuce juice so that the OD_600_ was 0.125, and Carbenicillin sodium was added to the solution (final concentration, 100 ug/mL). These are used as a recovery *E. coli* suspension.

#### 2.17.5. Induction of Recovery from the O^3^Pc-MSP Knockdown Effect was Performed as Follows

The suspensions of 5-KD, 3-KD, and control *E. coli* were washed three times with K-DS and left to swim in K-DS overnight to release the food vacuole contents within the *Paramecium* cells. After that, a recovery *E. coli* suspension was added to these *Paramecium* suspensions to a cell density of 400 cells/mL. The mixture was then incubated at 25 °C. The cells were cultured for about 45 hours. 5-KD was fed with a recovery *E. coli* suspension prepared using the 5′-L4440 vector-transformed HT115 strain. From now on, this will be referred to as 5KD-R (Rescue 5-KD from knockdown). Similarly, 3-KD was fed a recovery *E. coli* suspension from the 3′-L4440 vector-transformed HT115 strain, and the control was fed a recovery *E. coli* suspension from the L4440 vector-transformed HT115 strain. They were called 3KD-R (Rescue 3-KD from knockdown) and cont-R. To verify whether the Pc-MSP knockdown effect was reversed, we conducted semi-quantitative RT-PCR to measure the expression level of Pc-MSP mRNA utilizing 5KD-R, 3KD-R, and cont-R. Furthermore, we compared the intracellular localization of O^3^Pc-MSP using an indirect fluorescent antibody method. Moreover, we calculated the percentage of cells that expressed mating reaction activity in the knockdown strain using the method described elsewhere.

### 2.18. Detection of O^3^Pc-MSP mRNA by Semi-Quantitative RT-PCR Method and Image Analysis

*Paramecium* cells expressing mating reaction activity and cells not expressing mating reactivity from both mating types were washed three times with K-DS. Then, 50 cells were randomly selected, and semi-quantitative RT-PCR was performed to detect O^3^Pc-MSP mRNA expression. After agarose electrophoresis, the nucleotide bands were stained with ethidium bromide to identify the O^3^Pc-MSP mRNA band. The brightness of the band was measured using the NIS element (Nikon, Japan). The amount of O^3^Pc-MSP mRNA in each sample was standardized using α-tubulin.

### 2.19. Process of Preparing a Sample for the Microinjection of the O^3^Pc-MSP PcVenus Vector

A *P. caudatum* expression vector pTT3 Pc-MSP-PcVenus 7.0 carrying a PcMSP-PcVenus fusion gene was constructed using the pTT3H2B-pcVenus expression vector [31]. pTT3 PcMSP-PcVenus 7.0 was made by replacing the immaturin site of pTubMcsPcVenus-Immaturin with the O^3^Pc-PMS gene. Pc-MSP in the vector diagram indicates O^3^PC-MSP. After cloning and mass production of this vector, it was linearized with the restriction enzyme *Bam*HI and used as a sample for microinjection (about 1.0 μg/μL). Recipient cells of *Paramecium* were cultured on depression slides. Well-fed cells were isolated, and approximately 10 pL of the vector was injected into the recipient macronucleus. After standing for about 30 min, the recipients were transferred to a fresh lettuce juice medium and incubated at 25 °C. Twenty-four hours after injection, transformants were detected by fluorescence emitted from PcVenus. We used the Hp1-YFP vector lacking the O^3^ Pc-MSP gene and O^3^ cell DNA as controls for microinjection.

The microinjection was performed using Haga’s two-needle method [32], an improved version of Koizumi’s single-needle method. The transformation was induced by injecting approximately 5-10 pL of a solution containing the plasmid into the macronuclei of the recipient cells.

## 3. Results

### 3.1. Complete Base Sequence of O^3^Pc-MSP Gene

In order to verify that O^3^Pc-MSP is an O^3^-mating substance, a candidate polypeptide was extracted from the ciliary membrane fraction of *Paramecium* as a first step. Then, this polypeptide’s partial amino acid sequence was determined by TOF-MS. The DNA base sequence was estimated from the amino acid sequence, and primers used for PCR were designed and synthesized artificially. The region encoding O^3^PcMSP in the genomic DNA was amplified using these primers. The reason for gene cloning is that it is more convenient to estimate amino acid sequences from genomic DNA base sequences than to determine amino acid sequences directly from the polypeptides in question. The PCR products from *Paramecium* O^3^ and E^3^ cells were sequenced, and the complete nucleotide sequences were determined (Figure 1a). The O^3^Pc-MSP gene is composed of 1618 bases, including seven introns ranging from 20 to 23 bases. The homolog gene in the E^3^ mating type is also 1618 bases long and contains seven intron regions of 20–23 bases in length. Upon comparing the sequences of O^3^ and E^3^ types, it was discovered that there were single nucleotide substitutions at nine locations: eight in the exon region and one in the intron region. These substitutions are indicated in red letters in Figure 1a. A BLAST search of the nucleotide sequences for both O^3^ and E^3^ nucleotide sequences did not reveal any completely matching gene sequences, thus indicating that these are new genes.

### 3.2. Deduced Complete Amino Acid Sequences of Pc-MSP O^3^ and E^3^

The exon region of the O^3^Pc-MSP gene was translated into an amino acid sequence using the ORF finder website (Figure 1b). The O^3^Pc-MSP comprised 489 amino acids and had all three types of partial amino acid sequences obtained by mass spectrometry. When compared to the E^3^ type, which had single nucleotide substitutions in nine locations in their DNA base sequences, it was found that the E^3^ type had the same amino acid sequence. According to the results obtained from Resource Portal, the estimated molecular weight of the substance was 52.67 kDa.

A BLAST search to predict the domain structure of the O^3^Pc-MSP showed a protein kinase C-like domain in the 46th to 304th amino acid sequence region. Additionally, four consecutive EF-hand motifs, which are calcium ion binding regions, were identified between the 350th and the C-terminal amino acids.

### 3.3. Molecular Phylogeny of O^3^Pc-MSP

Initially, we employed the NCBI domain search (https://www.ncbi.nlm.nih.gov/structure/cdd/wrpsb.cgi, accessed on 5 January 2024), using the full-length amino acid sequence of O^3^Pc-MSP (AHW46386) as the query, to investigate the potential function of O^3^Pc-MSP. The search results revealed that O^3^Pc-MSP comprises a catalytic domain of serine/threonine kinase at the N-terminal side and an EF-hand motif for calcium binding at the C-terminal side. These findings suggest that O^3^Pc-MSP functions as a calcium-dependent protein kinase (Figure 1b). Additionally, the search identified specific amino acid residues responsible for ATP- and Ca^2+^-binding (Figure 1b).

A BLAST search with the amino acid sequence of O^3^Pc-MSP revealed similarities to four proteins from *Paramecium*; however, these protein sequences lacked annotations. To expand our search, we explored homologous proteins in closely related ciliate genera, including *Tetrahymena* and the parasitic ciliates *Ichthyophthirius* and *Pseudocohnilembus*. A protein from *Tetrahymena thermophila* (XP 001022919) exhibited conserved domains characteristic of calcium-dependent protein kinases, including an EF-hand motif and an ATP binding site. Furthermore, amino acid sequence analysis of a homologous protein in *Pseudocohnilembus persalinus* (KRX00690.1) suggested a Ca^2+^/calmodulin-dependent protein kinase-like function. These findings strongly support the classification of O^3^Pc-MSP as a calcium-dependent protein kinase.

Finally, we performed multiple amino acid sequence alignments of O^3^Pc-MSP with homologous proteins, including those from *S. cerevisiae*, *D. melanogaster*, *C. elegans*, *M. musculus*, and *H. sapiens*, revealed highly conserved amino acid residues that are diversely shared among these homologues (Appendix A). The ML (maximum likelihood) analysis of O^3^Pc-MSP, along with 14 homologues, revealed that the 10 ciliate proteins formed a distinct monophyletic group, termed clade 1 (numbers in square boxes in the phylogenetic tree, Figure 1d), which further diverged into two separate monophyletic clades 2 and 3 (Figure 1d). Clade 2 subsequently split into two additional monophyletic clades, 4 and 5, both supported by strong statistical evidence. Clade 4 exclusively comprised *Paramecium* proteins; however, O^3^Pc-MSP was positional as a sister taxon to the other four *Paramecium* proteins. This suggests that the amino acid sequence of O^3^Pc-MSP is comparatively unique relative to the other *Paramecium* homologues included in this study.

### 3.4. Detection of the O^3^Pc-MSP mRNA by RT-PCR

To confirm the transcription of the O^3^Pc-MSP gene into mRNA, we conducted RT-PCR to amplify the entire length of the O^3^Pc-MSP mRNA. When RNA derived from the O3 type was used as a template (Figure 2a, lane O3), a band of approximately 1.5 kb was detected. A similar-sized band was also detected when RNA from the E^3^ type was used (Figure 2a, lane E^3^).

### 3.5. Detection of the O^3^Pc-MSP from Ciliary Membrane Fraction by Western Blotting

Western blotting using anti-MSP antibody was performed using O^3^ and E^3^ type of ciliary membrane fractions expressing mating reactivity. A band with a molecular weight of approximately 52 kDa was detected in the fraction derived from the O^3^ type (Figure 2b, lane C). In contrast, no band was detected in the fraction derived from the E^3^ type (Figure 3, lane D). No band was detected in the control, where only the secondary antibody was used, regardless of the mating type (Figure 2b, lanes A and B).

### 3.6. Detection of O^3^Pc-MSP mRNA by Semi-Quantitative RT-PCR

To identify mRNA in mating reactive cells, total RNA was extracted from 50 cells collected from both mating reactive and nonreactive O^3^ types. Semi-quantitative RT-PCR was performed using these total RNAs as templates (Figure 2c). The mRNA of the O^3^Pc-MSP was detected in cells expressing mating reactivity (Figure 2c, upper row, lane A) and not in cells that did not express mating reactivity (Figure 2c, upper row, lane B). α-tubulin mRNA was detected to the same extent in cells expressing mating reactivity and in cells not expressing it (Figure 2c, bottom, rows A and B).

### 3.7. Indirect Fluorescence Image Using an Anti-MSP Antibody dsuring Vegetative Growth and the Initial Mating Process

We examined the localization of O^3^Pc-MSP in cells expressing mating reactivity during vegetative growth that had formed aggregates during the mating reaction (Figure 3a). DIC images confirmed the presence of cilia all over the cell surface (Figure 3a, left). In fluorescence images using anti-MSP antibodies, strong O^3^Pc-MSP signals were detected from the cilia in the ventral region from the front end of the cell to the vicinity of the oral apparatus (Figure 3b, left). On the other hand, no fluorescent signal appeared in the cilia of cells in which mating reactivity was not expressed (Figure 3c). Upon comparing the fluorescence intensity of cilia, we observed a significant increase in the intensity of cells expressing mating reactivity compared to those that did not (Figure 3d). There was a significant difference between the cells that formed aggregates due to the mating reaction and not expressing mating reactivity (Student’s *t*-tests, *p* < 0.001).

An investigation was conducted to determine changes in fluorescence intensity during the conjugation process. The photo on the left shows cells that have initiated the mating process (Figure 4a). A strong fluorescent signal was observed in the cilia from the head to the oral apparatus, indicating the presence of O^3^Pc-MSP. Cells in the hold-fast union stage were selected from samples taken 45 min after the start of the mating reaction. Fluorescence images showed a weak signal in the cytoplasm, but no signal was detected from the cilia. After 90 min of the mating reaction, a paroral union stage was observed, and a weak signal was detected throughout the cytoplasm, but no signal was detected from the cilia. We measured the relative fluorescence intensity around the cell membrane of the three groups (Figure 4b). There was a significant difference between cells that formed aggregates due to the mating reaction and the other two groups (*p* < 0.001, Tukey’s multiple comparison test, *n* = 20).

### 3.8. Verification of O^3^Pc-MSP mRNA Knockdown Effect by Semi-Quantitative RT-PCR

Three experiments were performed using feeding RNA interference to confirm the correlation between the amount of O^3^Pc-MSP mRNA and the amount of O^3^Pc-MSP in the ciliary membrane. Two O^3^Pc-MSP knockdown strains, 5-KD and 3-KD, were investigated. The control experiment used the *E. coli* HT115 strain transformed with the L4440 vector. When we verified the amount of O^3^Pc-MSP mRNA in the 5-KD and 3-KD strains using semi-quantitative RT-PCR, O^3^Pc-MSP mRNA was present only in the control sample. α-Tubulin mRNA was detected with similar intensity in all three samples (Figure 5a). O^3^Pc-MSP localization was examined in knockdown strains using immunofluorescence. The control group revealed signals throughout the cytoplasm and cilia. Both 5-KD and 3-KD groups showed weaker signals in the cytoplasm and no signal from ventral cilia (Figure 5b). The fluorescence intensity surrounding the cell membrane was measured for control, 5-KD, and 3-KD strains (Figure 5c). There were significant differences between the control group and groups 5-KD and 3-KD. (Tukey’s multiple comparison tests, *p* < 0.001). The percentage of cells expressing mating reactivity is shown in Figure 5d. The percentage of cells expressing mating reactivity in the groups after knockdown treatment was significantly lower than in the control group (Tukey’s multiple comparison tests, *p* < 0.001). It was observed that O^3^Pc-MSP mRNA did not appear in cells that were treated with RNA interference. However, a fluorescent signal was detected in the cytoplasm. This could be due to the fact that RNA interference was 100% efficient. To improve the sensitivity of mRNA detection, quantitative RT-PCR, also known as real-time PCR, can be used instead of semi-quantitative RT-PCR. Real-time PCR is expected to improve the accuracy of RNA interference experiments.

### 3.9. Verification of O^3^Pc-MSP Knockdown Recovery Strains by Semi-Quantitative RT-PCR

We found that the knockdown effect led to a significant reduction in mating reactivity. Recovery experiments were performed using each knockdown strain to verify whether the mating reactivity of the knockdown cells would be restored. The first two strains recovered from knockdowns were 5KD-R and 3KD-R, respectively. The third strain was a control called cont-R. The study confirmed the recovery from the knockdown effect through semi-quantitative RT-PCR analysis. The O^3^Pc-MSP mRNA was found in all the cont-R, 5KD-R, and 3KD-R samples (the upper row of Figure 6a). The α-tubulin was expressed in all samples (cont-R, 5KD-R, and 3KD-R) (the bottom row of Figure 6a).

### 3.10. Indirect Fluorescence Image Using the Anti-MSP Antibody of O^3^Pc-MSP Knockdown Recovery Strain

An indirect immunofluorescence test was performed to verify the intracellular localization of O^3^Pc-MSP in the knockdown recovery strains. After testing the recovered strains using the same method as the knockdown strains, they appeared to have similar fluorescence properties. The fluorescent signals of O^3^ Pc-MSP were observed throughout the cytoplasm and around the oral apparatus, particularly in the front-end region of the cell in the cont-R, 5KD-R, and 3KD-R samples (Figure 6b). No fluorescent signal was detected in the fluorescent image of only the secondary antibody. The fluorescence intensities around the cell membranes of the cont-R, 5KD-R, and 3KD-R samples were observed and found no significant difference among these three groups using Tukey’s multiple comparison tests (Figure 6c). 

### 3.11. Percentage of Cells Expressing Mating Reactivity of O^3^Pc-MSP Knockdown Recovery Strain

The percentage of cells expressing mating reactivity was measured in three groups: O^3^Pc-MSP knockdown recovery lines 5KD-R, 3KD-R, and cont-R. The percentage was 70.0 ± 14.6 for cont-R, 71.1 ± 8.8 for 5KD-R, and 72.2 ± 8.8 for 3KD-R (Figure 6d). Statistical tests using Tukey’s multiple comparison test showed no significant difference among the three groups. None of the groups showed any mating reaction with the O^3^ type, which is the same mating type.

### 3.12. Comparison of O^3^Pc-MSP mRNA Levels between O^3^ and E^3^ Mating Types

Our research shows that cells expressing mating reactivity in both mating types contain O^3^Pc-MSP mRNA, while cells that do not express mating reactivity do not contain it (Figure 7a, Upper lanes O^3^ and E^3^). On the other hand, α-tubulin mRNA was detected in all cases (Figure 7a, lower three lanes MTS). By analyzing the normalized values of the O^3^Pc-MSP mRNA with the amount of α-tubulin mRNA (Figure 7b), we found that the O^3^ type expressing mating reactivity had approximately 1.5 times higher levels of O^3^Pc-MSP mRNA than the E^3^ type.

### 3.13. Comparison of the O^3^Pc-MSP Contained in the Ciliary Membrane Fraction between O^3^ and E^3^ Mating Types

We examined the correlation between the presence of O^3^Pc-MSP in membrane fractions and the expression of mating reactivity. The experiment compared the abundance of O^3^Pc-MSP in clones expressing mating reactivity to those that did not in both mating types. The findings indicate that only the O^3^-type ciliary membrane fraction expressing mating reactivity showed O^3^Pc-MSP detection after Western blotting of the ciliary membrane fractions extracted from these cells (Figure 7c).

### 3.14. Comparison of Subcellular Localization between Mating Types Using Indirect Immunofluorescence

No fluorescent signal was detected in the ventral cilia of E^3^ mating-type cells, even those expressing mating reactivity (Figure 8a) This indicates that O^3^Pc-MSP is present in the cytoplasm but not transported to the ciliary membrane. Further detailed experimental verification is required to confirm this finding in the future. Figure 8b summarizes the results of quantifying fluorescence intensity using the method explained in Figure 3a. Among E^3^ mating-reactive, O^3^ mating-nonreactive, and E^3^ mating-nonreactive, the O^3^ mating reactive showed the highest value (Figure 8b). Significant differences were observed between O^3^ mating-reactive and E^3^ mating-reactive, O^3^ mating-nonreactive, and E^3^ mating-nonreactive (*p* < 0.001).

### 3.15. Induction of Mating Type Changes in Different Mating Types or Syngen by Microinjecting the O^3^Pc-MSP Gene

To confirm the expression of mating type O^3^ in *Paramecium* cells, we developed an expression vector that integrates the O^3^Pc-MSP gene (Appendix A). The gene for fluorescent protein PcVenus is fused with the O^3^Pc-MSP gene by inserting it into the 3′ end. This results in the expression of both genes, causing the cell to emit green fluorescence when O^3^Pc-MSP is translated into protein. Three strains were selected as microinjection recipients: E^3^ from the same syngen and O type belonging to a different syngen (O^12^). Some cells emitting green fluorescence could form clones, even though the expression rate after microinjection was less than 10%. Clones with an O^3^ mating phenotype were generated from syngen E^3^ cells (Table 1). Clones exhibiting the O^3^ mating type have also been identified from syngen O^12^ cells. These results demonstrate that the O^3^Pc-MSP gene encodes a polypeptide responsible for O^3^-type mating substance. Groups of clones derived from control microinjections, Hp1-YFP vectors (without O^3^Pc-MSP gene), or DNA extracted from O^3^ cells did not exhibit O^3^ mating phenotypes.

## 4. Discussion

We have discovered new evidence that was not considered in our original hypothesis, so we will discuss these points first. An NCBI Blast search revealed that O^3^Pc-MSP is composed of three units. The first part of the amino acid sequence from the N-terminus has been previously described, but its function has yet to be predicted. The second part is the C-kinase domain, which is highly conserved among different species, from *Tetrahymena* to *Homo sapiens*. The last one is the EF-hand motif. Although the size of the EF-hand motif is very uniform in all species investigated, the nucleotide sequence is diverse. This finding suggests that the domains involved in ATP binding sites in O^3^Pc-MSP are highly conserved, but Ca-binding sites are diverse. However, no experimental information regarding the catalytic activity of O^3^Pc-MSP is available. In *Paramecium*, it has been reported that a polypeptide made from a base sequence highly homologous to glutathione transferase does not have this catalytic activity but has nuclease activity [34]. Although we have repeatedly conducted experiments to verify the kinase activity of O^3^Pc-MSP, we have yet to reach a clear conclusion. On the other hand, our unpublished observation suggests that calcium ions are involved in the entire mating process, from initiation to progression.

The second unexpected finding is that the O^3^Pc-MSP gene is also transcribed in E^3^ cells. However, the polypeptide of O^3^Pc-MSP is not detected by anti-O^3^Pc-MSP antibodies in E^3^ cells. Gene transfer experiments show that when the O^3^Pc-MSP gene is introduced into E^3^ cells, the transformants express mating type O^3^, suggesting that transcription and translation of the O^3^Pc-MSP gene are possible in E^3^ cells. It is necessary to consider an experimental system to confirm whether O^3^ mRNA translation is suppressed in E^3^ cells in natural conditions. In *the P. aurelia* complex, mating-type O is the default state [35,36,37], and additional genetic functions are needed for cells to become E-mating types. Complementation tests in *P. tetraurelia* defined three unlinked loci (*mtA*, *mtB*, and *mtC*) required for the expression of E-type [37]. Recent advances showed that the *mtA* gene encodes an E-type specific transmembrane protein, and *mtB* is an *mtA*-specific transcription factor. No gene is required for type O expression [7]. In *P. tetraurelia*, the following are considered. O-type mating substances are synthesized in both mating types, but O-type substances that can bind *mtA* are synthesized by post-translational processing in O-type cells. We need to conduct an experiment to verify the structural genes, transcription factors, and translation suppression mechanisms of mating-type substances in order to produce complementary mating types in the *Paramecium* genus.

We will now discuss the nine single nucleotide substitutions observed in the O^3^Pc-MSP genes of the O^3^ and E^3^ mating types. The mating type of *Paramecium caudatum* is determined by a pair of alleles, *Mt* and *mt*, located at a single locus [8]. In this study, we demonstrated that the O*^3^*Pc-MSP gene encodes a protein that determines the O-type mating type substance. There are nine nucleotide substitutions present in the E*^3^* locus. These substitutions, however, do not have any observable effects on the amino acid sequence. There are three possibilities: (1) The substitution is due to an error in the DNA base sequence analysis. (2) It could be due to individual variation and not specific to the mating type. (3) It is a single nucleotide substitution related to the mating type and has mating-type specificity. The O^3^Pc-MSP gene underwent analysis through five independent nucleotide sequence analyses, and all five results showed that single nucleotide substitutions occurred in the same manner among different mating types. This demonstrates the reproducibility of the results and makes it unlikely that any errors occurred during the sequencing process. In regards to points (2) and (3), Stock KYC01D7 (O^3^) has the same mating type as strain TAZ0460 (O^3^ type), and KYC01E4 (E^3^) has the same mating type as TAZ0462 (E^3^ type). The nucleotide sequences of these four strains were analyzed, and the results showed that two O^3^ types had the same base sequence. Similarly, the two E^3^ types also had the same base sequence. Based on this, the most likely possibility for the differences in the mating types is a single nucleotide substitution specific to the mating types.

Amino acid homology searches reveal a protein kinase C-like domain in the O^3^PC-MSP, a domain known to participate in fertilization in mammals. When a sperm attaches to the surface of a mammalian egg, it causes an increase in the concentration of calcium ions. This increase in calcium ions activates a protein kinase called PKC. Once activated, PKC moves towards the cell membrane and triggers the fusion of the egg and sperm cells [38,39]. In the unicellular eukaryote *Dictyostelium*, the ZYG1 protein has a phosphorylation site activated by PKC [40,41]. This activation induces cell fusion during sexual reproduction. When two *Paramecium* cells mate, their membranes partially fuse around the oral apparatus. This membrane fusion may be associated with the catalytic functions of mating-type substances during the two stages of mating pair: holdfast union and paroral union. These stages have been observed through immunofluorescence and are thought to involve transmembrane signal transduction mediated by calcium ions.

We investigated to establish a causal relationship between the O^3^Pc-MSP gene and the expression of mating reactivity. To achieve this, we used RNA interference. Our findings indicate a positive correlation between mRNA expression and the mating reactivity in O3-type *Paramecium*. Specifically, we observed that the mRNA of O^3^ Pc-MSP is always expressed in cells where mating reactivity is expressed. Furthermore, we found that when the expression of this gene is inhibited, the mating reactivity disappears. Conversely, when we removed the inhibition of the O^3^Pc-MSP gene expression, the mating reactivity reappeared. These observations led us to conclude that the Pc-MSP protein is essential for expressing mating reactivity in O^3^-type *Paramecium*.

We have constructed a vector to produce a fusion protein using a fluorescent protein called PcVenus and a target gene [31]. We generated a vector comprising O^3^PC-MSP and PcVenus genes. This vector was injected into E-type cells from the same syngen or O-type cells from a different syngen to create transformants. We confirmed the presence of clones expressing O^3^ mating type among clones derived from both recipients. These results suggest that the O^3^PC-MSP gene is a structural gene that codes for O^3^ mating type substance. We combined the result with RNA interference and demonstrated that the O^3^Pc-MSP base sequence fulfills the necessary and sufficient conditions for an O^3^ mating type gene.

The localization of E^3^-type Pc-MSP was verified using Western blotting and indirect immunofluorescence methods, but no evidence was found for the presence of cilia localized on the ventral side. There could be three potential reasons for this. Firstly, E^3^-type Pc-MSP may be present in cilia in such low abundance that direct and indirect fluorescent antibody methods cannot detect it. This possibility is supported by experimental results, which indicate that the amount of E^3^-type mRNA is lower in the E^3^ type than in the O^3^ type. The second possibility is that O^3^Pc-MSP is not produced in the E^3^ type due to the presence of silent mutations. Silent mutations do not change the amino acid sequence but have been demonstrated to affect protein function post-translationally. Kimchi-Sarfaty et al. conducted an experiment to introduce a silent mutation into the MDR1 gene, a transporter protein in *E. coli* [42]. They found that such mutations affected the efficiency of protein synthesis. This occurs when the mutated codon becomes rare and not commonly used. As a result, the corresponding tRNA concentration is low, and the ribosomal protein translation rate is slower. This reduces protein synthesis or stops translation, resulting in abnormal three-dimensional structures in the polypeptide.

Nine mutations within the 310–458 amino acid sequence of the EF-hand motif region of O^3^Pc-MSP are silent and account for 6% of the 148 amino acids. The first base of the triplet was substituted with the first silent mutation at 148 amino acids, the second base replaced the seventh, and all other substitutions were found at the third base among the nine silent mutations. When a substitution occurs at the third base of the triplet, a change occurs at the first base on the antisense side. If translation were to occur from the antisense side, this change would create a protein with a new amino acid sequence. Since it has been revealed that O type and E type strains share the O^3^Pc-MSP gene, we consider the possibility that these two phenotypes, O type and E type, can be created by one structural gene. This can be explained by assuming that when the nucleotide sequence on the sense side is translated, the cell becomes type O, and when the nucleotide sequence on the antisense side is translated, the cell becomes type E.

A study on ciliary membrane proteins in *Paramecium* indicated the existence of membrane transport proteins [29], but their functions still need to be fully understood. There are currently no reports directly confirming the membrane transport efficiency of O^3^Pc-MSP. Various antibodies that can detect antigens in the ventral cilia are found in different species of the *Paramecium* genus [43,44]. To comprehend the binding mechanism between complementary mating types, understanding the O^3^Pc-MSP protein’s molecular structure is essential. Obtaining this knowledge will be a significant breakthrough in this field.

Some ciliates have mechanisms to prevent them from mating with individuals belonging to the same sibling group. Mating-type determination mechanisms in such species have developed in various ways, creating extremely large genetic diversity in closely related species [7,45]. Autogamy and selfing conjugation are classified as inbreeding and also play a crucial role in some ciliates. The balance between inbreeding and outbreeding regulation within a species, as well as the mechanisms that prevent interspecific hybridization, are still a topic of debate in terms of their evolutionary effects.

## Figures and Tables

**Figure 1 microorganisms-12-00588-f001:**
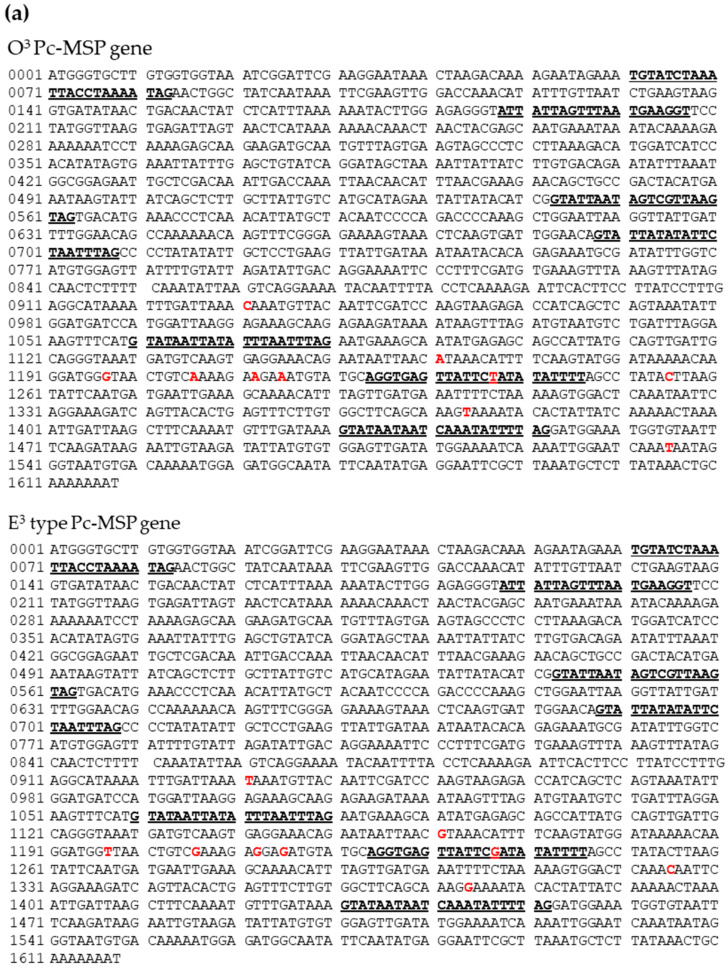
The complete base sequence of mating-type substance O^3^ and the homologous gene amplified from the E^3^ genome. (**a**) The DNA sequence contains 1620 bases with seven introns, as indicated in bold and underlined. Bases that differ between O^3^ and E^3^ are marked in red (nine locations). (**b**) The deduced amino acid sequence of the O^3^ mating type substance (489 amino acids). The area inside the orange box is very similar to the domain of the protein kinase C-like. The blue boxes highlight the EF-hand motifs, representing the regions that bind to calcium ions. (**c**) A three-dimensional ribbon model of O^3^Pc-MSP is displayed. The N-terminus of the O^3^Pc-MSP polypeptide is shown in blue, and the C-terminus in red. The ribbon model was produced by the computer graphic software Phyre 2 (protein homology/analogy recognition engine v. 2.0) [33]. The total shape deduced from the amino acid sequence of O^3^Pc-MSP is shown. (**d**) ML phylogeny of O^3^Pc-MSP and homologous proteins based on the full-length amino acid sequence alignment. The NCBI accession number for each protein is listed next to the respective species name. Numbers on the nodes indicate bootstrap values greater than 50% from 1000 replicates. The scale bar represents evolutionary distance, measured in units of amino acid substitutions per site. Clade numbers, as referenced in the Results section, are displayed as numbers inside squares above each node.

**Figure 2 microorganisms-12-00588-f002:**
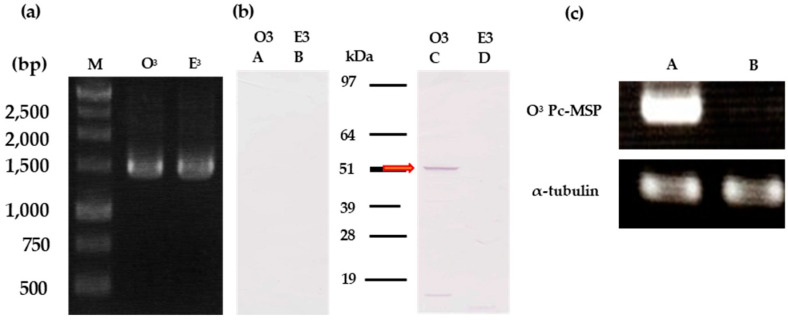
(**a**) Agarose gel electrophoresis image of O^3^Pc-MSP mRNA amplified by RT-PCR. RT-PCR was performed using total RNA extracted from O^3^ and E^3^ cells expressing mating reactivity as a template. The lanes represent the DNA size marker (M), O^3^ product (O^3^), and E^3^ product (E^3^). (**b**) Western blot images using anti-MSP antibody of ciliary membrane fractions prepared from O^3^ and E^3^ mating types that expressed mating reactivity. The left image shows the secondary antibody only, while the right image shows both the anti-MSP and secondary antibodies. Each lane of the image contains 50 µg of proteins, and the molecular weights obtained from the molecular weight markers are shown in the center of the figure. The red arrow indicates the band’s position, which was detected with an anti-MSP antibody from the ciliary membrane fraction of O^3^. The ciliary membrane fraction of O^3^ expressing mating reactivity is shown in (A) and (C), while the ciliary membrane fraction of E^3^ expressing mating reactivity is shown in (B) and (D). (**c**) Agarose gel electrophoresis image of O^3^Pc-MSP mRNA amplified by RT-PCR. RT-PCR was performed using total RNA extracted from O^3^ and E^3^ cells expressing mating reactivity as a template. The lanes represent the DNA size marker (M), the O^3^ product of mating reactive cells (A), and the O^3^ product of mating nonreactive cells (B).

**Figure 3 microorganisms-12-00588-f003:**
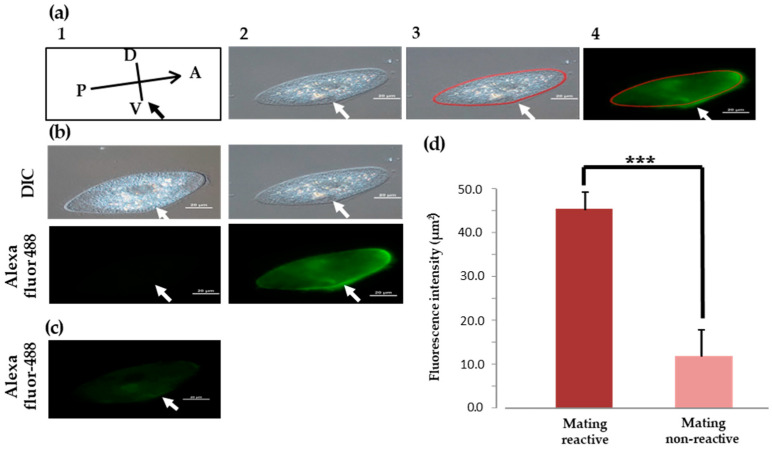
Indirect fluorescence image of an O^3^ cell expressing mating reactivity using anti-MSP antibody. White and black arrows in the photographs indicate the place of the oral apparatus. All of the scale bars displayed in the images represent a length of 20 micrometers. (**a**) The following is a method for measuring fluorescence intensity around cell membranes using indirect immunofluorescence. Panel 1 illustrates the cell’s head (A), tail (P), back (D), and abdomen (V), as well as the oral region indicated by arrows. Photo 2 displays the DIC image. Photo 3 displays 60 spots, each 200 μm² in size, outlining the cell in Photo 2. The location where the fluorescence intensity of the cell part is measured is indicated. Photo 4 superimposes the marked spots on the fluorescence image, measuring the fluorescence intensity within each spot. Data for graphing is obtained by converting the fluorescence intensity at each measurement point into a value per unit area (μm^2^). (**b**) These are the photographs of O^3^ cells that show mating reactivity. The upper left image is taken using a differential interference contrast microscope (DIC), while the lower left image shows the cell using only secondary antibodies, without primary antibodies. The upper right image is a DIC image, and the lower right image shows the localization of O^3^Pc-MSP using an anti-O^3^Pc-MSP antibody and a secondary antibody. The white arrows indicate the position of the oral appliance. (**c**) This is an image of non-mating reactive O^3^ cells stained with an anti-O^3^Pc-MSP antibody and a secondary fluorescent antibody, with no detected fluorescence signal. (**d**) The graph shows the comparison of the intensity of indirect immunofluorescence between O^3^ cells that express mating reactivity and those that do not. The vertical axis of the graph shows the relative fluorescence intensity. The graph displays the mean and standard deviation of 10 cells, and three stars in the graph indicate that a significance test was conducted, with a *p*-value of less than 0.001 (Student’s *t*-test).

**Figure 4 microorganisms-12-00588-f004:**
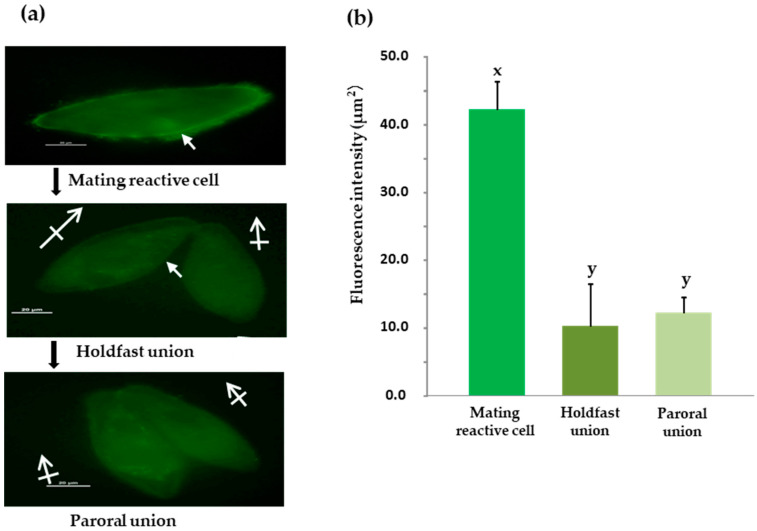
Changes in O^3^Pc-MSP localization during the conjugation process. (**a**) The images are visualized through an indirect immunofluorescence method. The photo at the top displays cells that are expressing mating reactivity prior to the mating reaction. The center shows the holdfast union, and the bottom shows the paroral union. White arrows indicate the position of the oral apparatus. The scale bar shown in the photo represents 20 micrometers. The scale bars displayed in the images represent a length of 20 micrometers. (**b**) The graph shows the quantified fluorescence intensity of each cell. The vertical axis is relative intensity. The graph displays the average value and standard deviation of the cells. Tukey’s multiple comparison test reveals that there is a statistically significant difference between the alphabets x and y (*n* = 20, *p* < 0.001).

**Figure 5 microorganisms-12-00588-f005:**
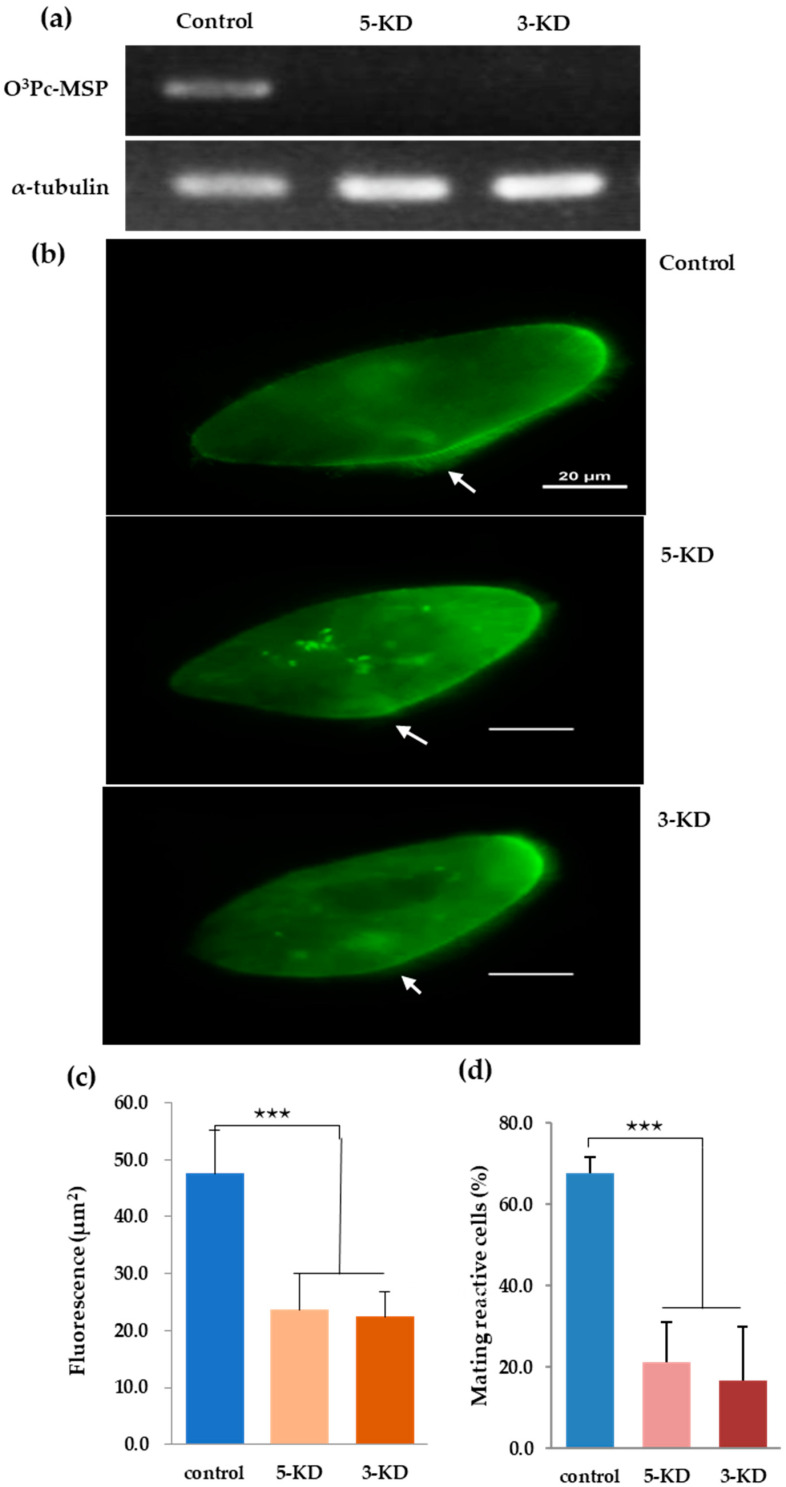
Semi-quantitative RT-PCR using total RNA in a knockdown strain of the mating type substance gene. (**a**) Total RNA was extracted from a strain with a knocked-down mating-type substance gene. RT-PCR was performed, and agarose electrophoresis images of RT-PCR product DNA of 5′-KD, 3′-KD, and control are shown. (**b**) Indirect fluorescence images of control, 5′-KD knockdown strain, and 3′-KD knockdown strain prepared by feeding RNA interference method. White arrows indicate the oral region. The scale bars shown in the photo represent 20 micrometers. (**c**) The graph shows the quantified fluorescence intensity of each cell. The vertical axis is relative fluorescence intensity. The graph displays the average value and standard deviation of the ten cells. (**d**) The percentage of cells expressing mating reactivity. Tukey’s multiple comparison test shows a statistically significant difference between the control group and the treated groups. Three stars indicate that the risk level is smaller than 0.001 (*p* < 0.001).

**Figure 6 microorganisms-12-00588-f006:**
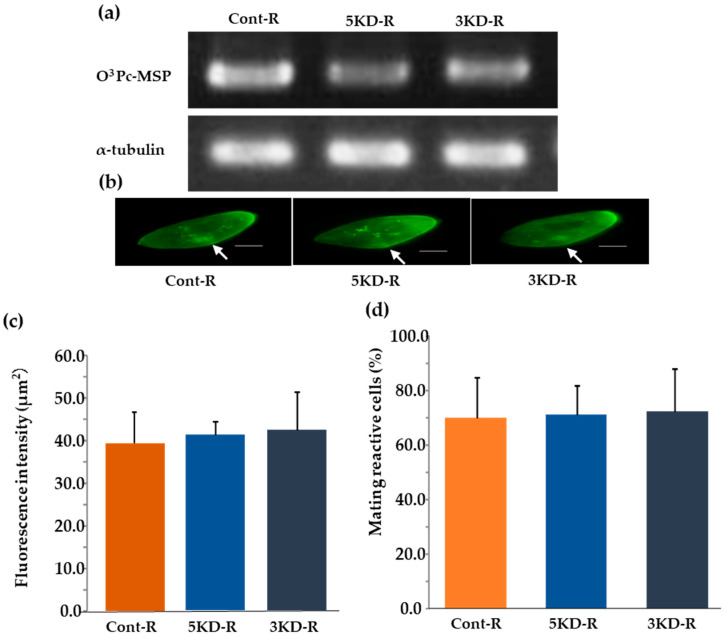
Semi-quantitative RT-PCR image of mRNA for mating type substance using total RNA from knockdown recovery strain. (**a**) Total RNA was extracted from 5KD-R and 3KD-R, knockdown recovery strains of the mating type substance gene, and cont-R, a control group. Then, semi-quantitative RT-PCR was performed using primers to amplify the mRNA of the mating type substance. (**b**) A fluorescent image of the knockdown recovery strains obtained through the indirect immunofluorescence method is displayed. Photos show cont-R, 5KD-R, and 3KD-R, respectively, from left to right. White arrows indicate the oral region. The scale bars shown in the photos represents 20 micrometers. (**c**) The graph displays the fluorescence intensity measured from indirect fluorescence images of the knockdown recovery strain around the cell membrane. The vertical axis indicates the relative fluorescence intensity. Tukey’s multiple comparison tests found no significant difference among the three groups (*p* < 0.05). (**d**) The graph shows the average percentage of mating reactive cells in the clone of knockdown recovery strains. The vertical axis is the percent of mating reactive cells.

**Figure 7 microorganisms-12-00588-f007:**
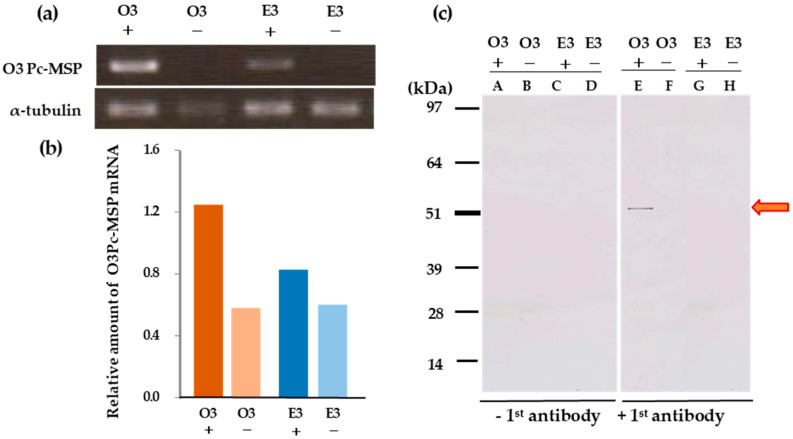
Comparison of O^3^ type and E^3^ type regarding O^3^Pc-MSP gene expression. (**a**) Semi-quantitative RT-PCR using total RNA from cells expressing or not expressing mating reactivity in both mating types. A band indicating the presence of O^3^Pc-MSP gene mRNA was detected in O^3^ and E^3^ cells expressing mating reactivity, but no band was detected in cells not expressing mating reactivity in both mating types. (**b**) The graph shows the results of normalizing the brightness of the O^3^Pc-MSP DNA band detected in photo A with the brightness of the α-tubulin band. +: mating reactive cell, −: mating nonreactive cell. (**c**) Detection of O^3^Pc-MSP polypeptide in the ciliary membrane fraction of E^3^ cells expressing mating reactivity by Western blotting. Each lane contains 50 µg of protein. The red arrow indicates the position of the molecular weight of the O^3^Pc-MSP protein (approximately 52 kDa). The O^3^Pc-MSP band was detected only in the O^3^ cilia fraction with mating reactive.

**Figure 8 microorganisms-12-00588-f008:**
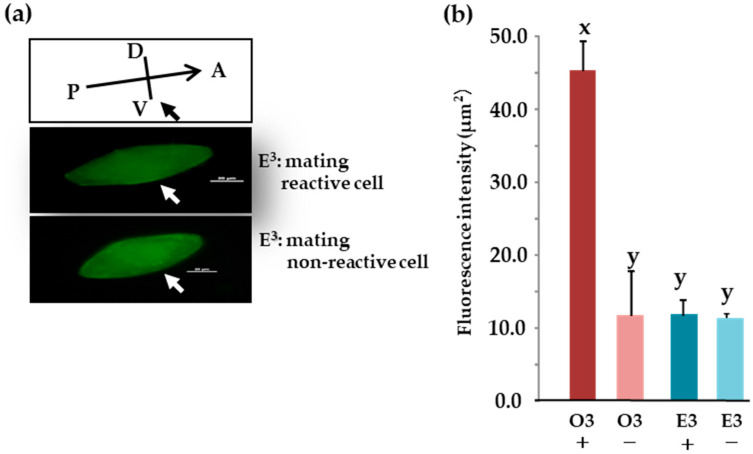
Localization of the O^3^Pc-MSP polypeptide visualized with anti-MSP antibody. (**a**) Indirect immunofluorescence imaging using anti-MSP antibody in cells expressing or not expressing mating reactivity in E^3^ cells. The fluorescence intensity in the cytoplasm was increased by enhancing the brightness of the photographs on a computer. However, no amplification effect was observed in the cilia. White and black arrows indicate the oral region. The scale bar shown in the photo represents 20 micrometers. (**b**) The graph shows fluorescence intensity near the cell membrane in images from immunofluorescence. The vertical axis is relative fluorescence intensity. +: mating reactive cell, −: mating nonreactive cell. Tukey’s multiple comparison test reveals that there is a statistically significant difference between the alphabets x and y (*n* = 10, *p* < 0.01).

**Table 1 microorganisms-12-00588-t001:** Mating type change in cells of different syngen by microinjecting the O^3^ Pc-MSP gene.

Stock Name	Mating Type	Mating Type after Microinjection
Bw 15-3	E^3^	O^3^
Ai212	O^12^	O^3^
Mkwp	E^3^, E^12^	E^3^, E^12^

At least 50 cells were microinjected for each stock. The following day, only about ten cells emitting green fluorescence were confirmed. After that, only one fluorescent clone could be established for each stock. In the case of Mkwp, it was determined that the transformation was of the transient type. In the control injection group, the injection of the Hp-1vector resulted in a permanent transformant that showed green, but there was no change in the mating type.

## Data Availability

Data are contained within the article.

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
