# Peer review of "Identification and Characterization of the Gene Responsible for the O^3^ Mating Type Substance in *Paramecium caudatum"

_microorganisms, 2024, doi:10.3390/microorganisms12030588_

Round 1

Reviewer 1 Report

Comments and Suggestions for Authors

In the manuscript, the authors identified O3Pc-MSP gene from Paramecium caudatum. It is widely present in eukaryotes and exhibits high homology among closely related species. O3Pc-MSP has nine silent mutations compared to the complementary mating type of the E3 homologue gene. O3Pc-MSP was found in the ciliary membranes and located from the head to the ventral side of cells. RNAi O3Pc-MSP showed O3Pc-MSP positively correlated with the expression of mating reactivity. The authors also found that mating type changes in different mating types or syngen by microinjecting the O3Pc-MSP.

1.       The title is ambiguous, the authors should focus on the main findings for O3Pc-MSP study.

2.       In the abstract, what is causal relationship between “O3-type substance” and “O3Pc-MSP”.

3.       In Line 483, “The PCR products from Paramecium O3 and E3 cells were sequenced, and the complete nucleotide sequences were determined”. What is PCR products and why the authors clone the gene? The authors should explain it.

4.       In Line 474-480, according to the results, “2.19 Molecular Phylogeny of O3Pc-MSP” should put in the ahead.

5.       Fig 2-Fig4 should reorganize into one picture or put them in the supplementary materials.

6.       In Fig 3, a band about 16KD was specifically occurred in the lane C, what is it?

7.       In Line 588, “not in cells that did not express mating reactivity (Figure 4, upper row, lane B)”, however, in line 594 “E3 product (B)”. Is this same product?

8.       In Fig 7 (b). 5-KD cell is bigger than control, is it new phenotype?

9.       Line 690-692,” The fluorescent signals of O3Pc-MSP were observed throughoutthe cytoplasm and around the oral apparatus, particularly in the front-end region of the cell in the cont-R, 5KD-R, and 3KD-R samples”, where is the oral apparatus, authors should clearly indicate it.

10.   Line 766-767, “The gene for fluorescent protein PcVenus is fused with the O3Pc-MSP gene by inserting it into the 3' end.” The localization picture of O3Pc-MS should show.

11.   In table1, the authors showed mating type change in cells of different syngen by microinjecting the O3 Pc-MSP gene. The O3 mating type cells after microinjection can mate with E3 cells?

12.   Fig 11 should move in supplementary materials.

Comments on the Quality of English Language

The active voice was overused:

For example: We conducted an investigation into changes in fluorescence intensity during the conjugation process. We selected cells at the hold-fast union stage from samples taken minutes after the start of the mating reaction. Fluorescence images showed weak signals in the cytoplasm, but no signal was detected from the cilia (Figure 6(a), middle). After minutes of the mating reaction, the paroral union stage was observed, and a weak signal was detected throughout the cytoplasm, while no signal was detected from the cilia (Fig ure 6(a), right). We measured the relative fluorescence intensity around the cell membrane of three groups (Figure 6(b)).

We conducted an indirect immunofl uorescence test to verify the intracellular localization of O 3 Pc-MSP in the knockdown recovery strains. After testing the recovered strainsusing the same method as the knockdown strains, we confirmed that they had similar fluorescence properties. The fluorescent signals of O 3 Pc-MSP were observed throughout the cytoplasm and around the oral apparatus, particularly in the front-end region of the cell in the cont-R, 5KD-R, and 3KD-R samples (Figure 8(b)). No fluorescent signal was detected in the fluorescent image of only the secondary antibody (data not shown). We also measured the fluorescence intensities around the cell membranes of the cont-R, 5KD-R, and 3KD-R samples and found no significant difference among these three groups using Tukey's multiple comparison tests (Figure 8(c))

Reviewer 2 Report

Comments and Suggestions for Authors

The reviewed paper dealt with identification of gene responsible for the mating type regulation in Paramecium caudatum. The research was carried out at a high methodological level. However, it is not entirely clear how the mechanism discovered by the authors ensures the reproductive isolation announced in the title of the article. As far as the reviewer knows, the mating type system in ciliates prevents self-conjugation, but does not ensure reproductive isolation. I recommend to clarify this point in the discussion.

Reviewer 3 Report

Comments and Suggestions for Authors

I regret to comment quite negatively on this ms by Chiba al for the following four major reasons. (i) The Authors claim (quoting from the title) to have identified, isolated and expressed via a heterologous system a “Paramecium mating type substance”, and regard this hypothetical substance as “a key molecule for reproductive isolation thought to be established in the early evolution of eukaryotes.” The data that have been presented are far from supporting that the substance in question: first, is specific to the two cell types that form the binary mating system of P. caudatum; second, is committed to effectively induce cell-cell unions in mating pairs; and third, is in some way functionally associated with early mechanisms of inter-species reproductive isolation. To my opinion, this substance in question is more likely identifiable with one of the many Ca++-dependent protein kinases that govern the ciliary activity. (ii) It is inexplicable (and puzzling) why the Authors make no reference at all to the mating-type genes, mating-type proteins and the mechanism of E and O mating-type inheritance that have originally been characterized by Sing et al (Nature, 2014) in P. caudatum-sister species, i.e. P. tetraurelia and P. septaurelia, and extensively reviewed by Orias et al (Ann Rev Microbiol, 2017). (iii) In relation to the key experimental steps involving DNA and RNA amplifications, no clear indication has been provided on which source protein the PCR primers have been designed. (iv) The general organization and articulation of the text is overloaded (Introduction and Discussion sections in particular) with a lot of out-of-context (and conceptually questionable) disquisitions, as well as with an amazingly lengthy list of detailed (and in large measure, useless) descriptions of experimental protocols that are of commonplace knowledge. (It seems to me that the ms is structured as a PhD thesis, rather than as a scientific article).

In conclusion, I feel confident to strongly recommend a complete restructuration of the ms in the light of what it is already well known in particular in Paramecium (and also in Tetrahymena, Blepharisma and Euplotes) about the mating-type genes and substances (pheromones).       

Comments on the Quality of English Language

Sentence construction and language style may be improved with the direct assistance of a native English

Reviewer 4 Report

Comments and Suggestions for Authors

See attachment.

Reviewer 5 Report

Comments and Suggestions for Authors

The Methodology is very extant and very specialised, which may not be so interesting for the common readers but it was for me. I like very much your attempt to combine molecular methods with an ecological approach, using "natural" feeding mechanisms to introduce information to the cells.

Even though the article is very long I recommend it to be published due to the lack of information on non-traditional methods of investigation. Many molecular biologists look to be enclosed in their own shells without direct communication with non-molecular methods. 

Round 2

Reviewer 1 Report

Comments and Suggestions for Authors

1. Line 723-727  O3Pc-MSP mRNA was present only in the control sample (Figure 5(a)).  O3Pc-MSP localization was examined in knockdown strains using immunofluorescence. The control group revealed signals throughout the cytoplasm and cilia. Both 5-KD and 3-KD groups showed weaker signals in the cytoplasm and no signal from ventral cilia (Figure 5(b)). The signal in cilia is not clear. The authors should enlarge it and show clearly ciliary localizaiton. In addition, O3Pc-MSP mRNA was disappeared, the O3Pc-MSP signal is still detected in the cytoplasm. Why?

2. Line832-833.  No fluorescent signal was detected in the ventral cilia of E3 mating-type cells, even those expressing mating reactivity (Fig8a) . But In Fig 8a, the signal is different in the  region arrows indicated.

Reviewer 3 Report

Comments and Suggestions for Authors

In revising their ms, the Authors have substantially maintained the idea/tenet that they have identified and characterized a ‘mating type substance’ from Paramecium caudatum. Sorry for my insistence in maintaining the idea that the substance in question is much more likely a kinase involved in Paramecium ciliary movement/regeneration. The Authors probably forget noticing that in mating paramecia, the cilia of the ventral surface are not only directly involved in making cells sticking/clumping with one another (before stably fusing in pairs); also, they completely change behavior/dynamics. In vegetative cells, ventral/circumoral cilia strongly beat to move cells. In cells interacting for mating, they ‘freeze’ their beating to become stiffer and instrumental to the cell-cell ‘ciliary adhesion’ (instrumental also to mutual ‘sex’ recognition?) in mating pairs. May or may not this mutated ciliary behavior account for quantitative variations in the expression of the identified substance? Furthermore, it seems to me that the widespread phylogenetic/homology relationships of the identified ‘mating type substance’ among macro-eukaryotes are much more consistent with a ciliary kinase rather than with a ‘sex’ substance that (at least in principle) should be characterized by a tight species-specificity. In any case, I do not wish insisting on my point. I am well aware that the responsibility of the ms content resides on Authors’ (and Editor’s) side.  Leaving apart the above major point, I have to frankly confess my expectation for a conceptually and formally more rigorous revision. Just a few examples of critical points that should be removed/improved.

(i) “Sexual reproduction” is a concept that applies to multicellular organisms, not to protists. See for example chapter 4 “Reproduction and Sex” in 1989 M Sleigh textbook “Protozoa and other protists”, or 1985 D Nanney textbook “Experimental Ciliatology”. In ciliate conjugation, there is no reproduction. Two cells start mating and two stop mating (in Vorticella, conjugation results in halving the cell number, from two to one!). Conjugation is a sexual process/phenomenon in the sense that there is a gene exchange via gamete-nuclei.

(ii) Without an ad hoc/anticipatory explanation, how many readers may understand that the “O” and “E” of the “O3/E3 mating type substance” and “O3/E3 Pc-MSP gene” stand for Odd and Even mating types of P. caudatum?

(iii) The following five sentences concluding the Introduction section are clearly emblematic for a very approximate/cursory (scientific and linguistic) revision. “We demonstrate the molecule and gene that distinguish individuals of the same species with complementary mating types. Our findings support the existence of molecules postulated by C.B. Metz as mating-type substances. We discussed findings related to the gene that defines the Odd-mating type of Paramecium belonging to Syngen 3 (O3Pc-MSP gene described in the study). We also discuss the feature of the catalytic domain deduced from the amino acid sequence of the mating type substance and the characteristics of the phylogenetic tree. We explored the characteristics and potential evolutionary roles of nine silent mutations that were revealed through comparison with homologous sequences of the complementary mating type E3.”

(iv) In writing that “The determination of mating type and inheritance pattern varies from species to species and is governed by species-specific molecules or rules.” and adding that “References for each species include ….” the Authors should note that the signaling molecules in question (nowadays more properly reported as ‘pheromones’) are ‘cell type-specific’ while “species-specific” are the protein-families that they form in the multiple mating systems, and that in Euplotes have been determined for their molecular/3D-structures not only in E. raikovi but also in E. nobilii and E. petzi (e.g. Alimenti et al EJP 2016, JEM 2022).

(v) Writing “Autogamy or selfing conjugation” is wrong, because autogamy is a self-fertilization (NOT “selfing conjugation”) between two gamete-nuclei that are generated by the same individual/cell, while conjugation involves (cross or self) fertilization in two cells.

Comments on the Quality of English Language

No
